# Disturbance of Vancomycin Infusion Flow during Multidrug Infusion: Influence on Endothelial Cell Toxicity

**DOI:** 10.3390/antibiotics11010016

**Published:** 2021-12-24

**Authors:** Maryline Drouet, Elodie Cuvelier, Feng Chai, Stéphanie Genay, Pascal Odou, Bertrand Décaudin

**Affiliations:** 1CHU Lille, ULR 7365–GRITA—Groupe de Recherche sur les Formes Injectables et les Technologies Associées, University Lille, F-59000 Lille, France; maryline.drouet@gmail.com (M.D.); stephanie.genay@univ-lille.fr (S.G.); pascal.odou@univ-lille.fr (P.O.); bertrand.decaudin@univ-lille.fr (B.D.); 2CHU Lille, Institut de Pharmacie, F-59000 Lille, France; 3Institut National de la Santé et de la Recherche Médicale (INSERM) U1008, University of Lille, UFR3S, 1 Place de Verdun, F-59000 Lille, France; feng.hildebrand@univ-lille.fr

**Keywords:** vancomycin, phlebitis, human umbilical vein endothelial cells, infusions, intravenous, simulation, in vitro techniques, toxicity tests

## Abstract

Background: Phlebitis is a common side effect of vancomycin peripheral intravenous (PIV) infusion. As only one PIV catheter is frequently used to deliver several drugs to hospitalized patients through the same Y-site, perturbation of the infusion flow by hydration or other IV medication may influence vancomycin exposure to endothelial cells and modulate toxicity. Methods: We assessed the toxicity of variations in vancomycin concentration induced by drug mass flow variations in human umbilical vein endothelial cells (HUVECs), simulating a 24 h multi-infusion therapy on the same line. Results were expressed as the percentage of viable cells compared with a 100% control, and the Kruskal–Wallis test was used to assess the toxicity of vancomycin. Results: Our results showed that variations in vancomycin concentration did not significantly influence local toxicity compared to a fixed concentration of vancomycin. Nevertheless, the loss of cell viability induced by mechanical trauma mimicking multidrug infusion could increase the risk of phlebitis. Conclusion: To ensure that vancomycin-induced phlebitis must have other causes than variation in drug mass flow, further in vitro experiments should be performed to limit mechanical stress to frequent culture medium change.

## 1. Introduction

Drug-associated phlebitis may be influenced by chemical and/or physical mechanisms. In particular, phlebitis is an adverse effect of vancomycin, as described in the literature [1,2], occurring at a frequency of 1–10% according to the product summary [3]. An Iranian study showed that the frequency of occurrence of phlebitis in children treated with vancomycin was dependent on the infusion modality (45.90 and 89.10%) [4]. This treatment causes concentration-dependent phlebitis: it appears to be less frequent when administered at a concentration below 5 mg/mL and better tolerated at a final concentration of 2 mg/mL [4,5]. In previous studies, we analyzed factors influencing vancomycin (VAN)-associated phlebitis in peripheral intravenous (PIV) infusion. We showed that vancomycin endothelial toxicity depended on concentration and infusion duration [6] and that vancomycin in combination with other antibiotics increased endothelial toxicity, enabling us to recommend using a separate line to infuse vancomycin via a PIV access [7]. However, in surgical and intensive care units, multiple simultaneous infusion of several drugs with one catheter (multi-infusion) could lead to dosing errors and highs and lows in medication levels [8,9]. Moreover, our studies showed unexpected flow variations in PIV infusion during multi-infusion therapy through a single intravenous (IV) access, inducing variations in drug mass flow [10,11]. Disturbance to the infusion flow by hydration or other IV medication could influence vancomycin exposure to endothelial cells and modulate local toxicity. Notably, an inadvertent bolus can appear when carrier flow is restored abruptly in the reservoir with a dead volume [9]. The aim of this study was to assess the toxicity of variations in vancomycin concentration under in vitro conditions simulating multi-infusion on the same line to determine whether vancomycin flow rate variations increase local endothelial toxicity.

## 2. Results

Our result showed that vancomycin maintained at a fixed concentration for 24 h in a culture medium of human umbilical vein endothelial cells (HUVECs) caused a concentration-dependent loss of cell viability of about 40% for 4 mg/mL, which was consistent with previous results [6]. A further loss of viability (20.0 ± 5.8%) was observed for vancomycin at variable concentrations compared to vancomycin at a fixed concentration without removing the culture medium (Figure 1). 

However, no significant difference was observed regarding the rate of cell death between condition 2 (fixed concentration of vancomycin with medium removal) and condition 3 (variable concentration of vancomycin) (Figure 1), signifying that the excess mortality observed was induced by cellular stress due to solution removal and the addition of fresh solution (Figure 1). 

## 3. Discussion

The availability of HUVECs to test drug solutions for intravenous compatibility is a valuable alternative to animal models, as has been demonstrated by several studies which have analyzed antibiotic compatibility and the inflammatory process on HUVECs [12,13]. In this study, we wanted to mimic a 10 h continuous infusion of vancomycin because continuous infusion of vancomycin is preferred in Europe. The cytoxicity analysis after several hours of incubation allows us to quantify the toxicity of variations (high and low) in vancomycin levels induced by multi-infusion with a peripheral catheter. Previous studies in our laboratory demonstrated that flow rate disturbance, which could be affected by catheter position, patient movement and fluid container height, can influence drug mass flow in patients who are treated with multiple infusions, inducing an increase or a decrease in drug concentration at the catheter egress [10,11]. We hypothesized that vancomycin bolus induced by drug mass flow variations simulating clinical use could induce an excess death rate of endothelial cells surrounding the catheter outlet, where the endothelial cells are subjected to a continuous flow of vancomycin with poor dilution, due to the weak blood flow in a peripheral vein of a patient in bed. However, our results showed that a variable concentration of vancomycin had no significant influence on toxicity compared to a fixed concentration of vancomycin with medium removal. This experiment did not induce excessive cytotoxicity beyond the cellular stress associated with frequent change of the culture medium. The limitations of this study suggest that other experiments more suitable for simulating multi-infusion need to be performed to answer this scientific question. Nevertheless, endothelial cell toxicity induced by a mechanical trauma due to multi-drug infusion, simulated in our model by medium removal and the addition of vancomycin at a constant concentration, could increase the risk of phlebitis, thus confirming our previous study [7]. This mechanical cell stress could mimic the flush induced by infusion of drugs at a high flow rate on previously damaged endothelial cells surrounding the catheter outlet. As vancomycin is frequently used in co-infusion with different drugs at different rates, it is essential to consider infusion as a whole. Curran et al. reported that the use of infusion pumps decreased the rate of phlebitis [14], and our previous study recommended using a specific access for vancomycin infusion to avoid potential drug incompatibilities. Flow rate variations induced by multi-drug infusion on the same line could be a supplementary argument. It would therefore be interesting to explore the endothelial toxicity of vancomycin administered at different rates by another method without medium removal. Testing cytotoxicity in real time by administering vancomycin at a variable concentration on cell biochips would avoid the cell stress associated with frequent medium change. On the other hand, a recent study showed that leachable materials released by infusion sets during infusion could induce local cytotoxicity [15]. Other theories remain to be explored, such as the influence of infusion set materials, in order to optimize methods of administering vancomycin by peripheral intravenous infusion (PIV).

## 4. Materials and Methods

### 4.1. Cell Culture

The drugs used, the cell culture method, and the cell vitality assessment were described in our previous study [6]. The HUVEC cell line was provided by Promocell GmbH, Heidelberg, Germany. Briefly, these cells were cultured with cell culture medium (endothelial cell growth medium enriched with endothelial cell growth SupplementMix, Promocell GmbH, Heidelberg, Germany) for 24 h to obtain an 80% confluent monolayer in a 96-well plate as an in vitro model of the vascular endothelium. 

### 4.2. Incubation with Vancomycin

#### 4.2.1. Methodology to Study Concentration-Dependent Cytotoxicity

As shown in a previous study [6], vancomycin infusion at concentrations over 5 mg/mL induced 50% endothelial cell death over 24 h. We therefore only applied concentrations between 1 and 4 mg/mL. 

The culture medium was removed from each well of the monolayer cells, the cells were rinsed with Phosphate-Buffered Saline 1X, then a vancomycin solution was added to the monolayer cells. 

The range of vancomycin concentrations (1–4 mg/mL) was prepared with the vancomycin powder (Mylan, France) reconstituted in NaC1 (0.9%) to obtain a concentration of 500 mg/10 mL and diluted with cell culture medium at 50/50 (*v*/*v*). 

#### 4.2.2. Methodology to Study Multi-Infusion-Dependent Cytotoxicity

Flow variations during multi-infusion therapy were simulated as follows: we set a sudden decrease in drug delivery by replacing vancomycin (VAN) with NaCl (0.9%) for 30 min or 1 h every 1 h 30, and a sudden increase by trebling vancomycin concentration on one occasion for 30 min (Table 1, VAN with concentration variations). A schematic representation of variations in vancomycin concentrations over 10 h for the variable concentration group is shown in Figure 2. 

To assess cell stress due to frequent medium removal from monolayer cells, we also set a control group, in which the addition of vancomycin was replaced by new vancomycin solution at the same concentration at each hourly change (Table 1, VAN at fixed concentration with medium removal). After a 24 h culture, the percentage of cell viability was compared to controls established with a fixed concentration solution of vancomycin with and without medium removal.

Excess cell death was calculated by comparing the cell death rate obtained with (1) vancomycin at a fixed concentration without medium removal and (2) vancomycin at a fixed concentration with medium removal.

### 4.3. Cytotoxicity Testing

After a 24 h culture with exposure to antibiotic solutions, cell reactions were evaluated by fluorometric assay with non-toxic AlamarBlue^®^ dye (Interchim, Montluçon, France), which is equivalent to the MTT (3-(4,5-dimethylthiazol-2-yl)-2,5-diphenyltetrazolium bromide) test to determine mammalian cell toxicity [16,17]. After 24 h of exposure, results were expressed as the percentage of cell viability compared to a control established with NaCl (0.9%) mixed with cell culture medium at 50/50 (*v*/*v*). Each test group was performed in triplicate.

### 4.4. Statistical Analysis

Non-parametric tests were used to compare percentages of HUVECs surviving with the null hypothesis that there was no difference between the experimental conditions assessed. The Kruskal–Wallis test was used to assess the toxicity of vancomycin in the three conditions. In the event of a significant *p*-value (<0.05), an analysis using the Conover and Iman method was performed to detect significant differences between couples of contact time. Each of these tests was performed with XLSTAT software version 2012.2.01 (Addinsoft, Paris, France).

## 5. Conclusions

Although the endothelial toxicity of vancomycin is known to be concentration-dependent, vancomycin did not induce excessive cytotoxicity beyond the cellular stress induced by the simulated multi-infusion conditions. These results can be explained by the cell stress-induced excess mortality associated with the frequent change of the culture medium. Vancomycin-induced phlebitis could have other causes than variation in drug mass flow. To ensure that vancomycin-induced phlebitis must have other causes than variation in drug mass flow rate, further in vitro experiments should be performed to limit mechanical stress due to frequent culture medium change. If our results are confirmed, the mechanical stress related to multidrug infusion should be avoided by using a specific access for vancomycin infusion.

## Figures and Tables

**Figure 1 antibiotics-11-00016-f001:**
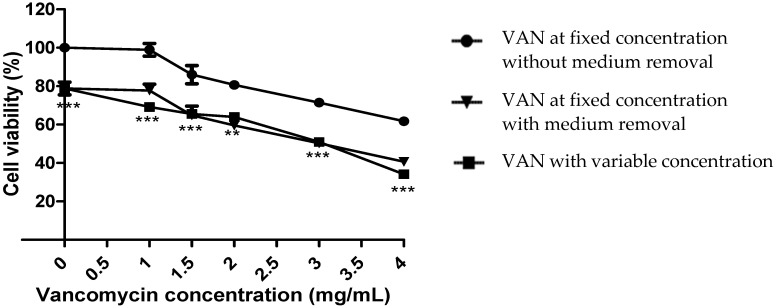
Cell viability of HUVECs after 24 h contact with vancomycin. Vancomycin (VAN) concentration ranged from 1 mg/mL to 4 mg/mL. ** *p* < 0.01, *** *p* < 0.001, for the comparison between VAN at a fixed concentration without medium removal and VAN with variable concentration. n = 3, error bars represent the standard error of the mean.

**Figure 2 antibiotics-11-00016-f002:**
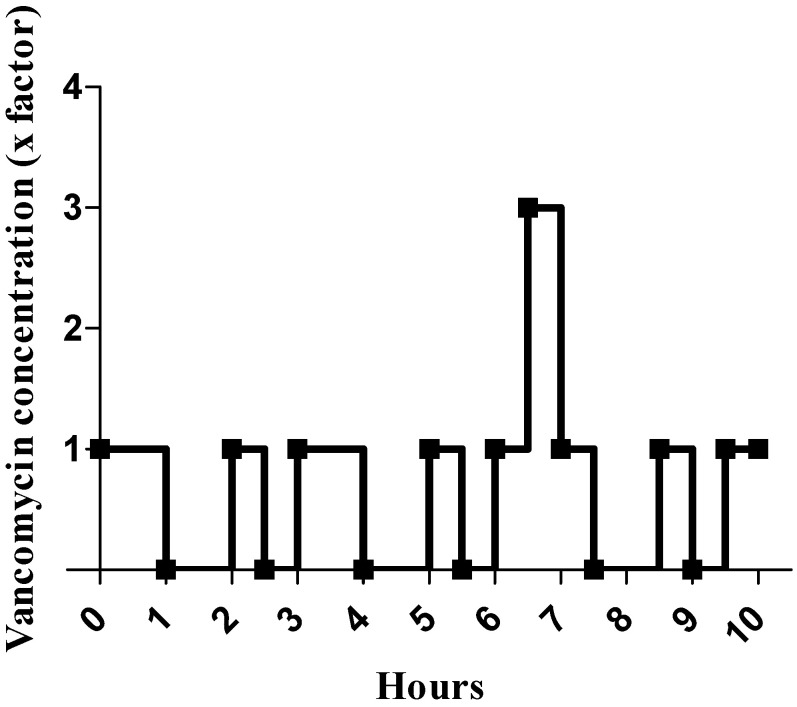
Schematic representation of vancomycin (VAN) concentration variations during the cellular test. Vancomycin concentration is represented by a multiple of the initial concentration, which is 1 mg/mL.

**Table 1 antibiotics-11-00016-t001:** Procedure for cellular test with variations in vancomycin (VAN) concentrations. HUVECs were exposed to vancomycin at a fixed concentration 1× (corresponding to 1, 1.5, 2, 3, and 4 mg/mL) with or without medium removal, or to vancomycin with a variable concentration, simulating clinical use over 10 h (H). NaCl (0.9%) replaced vancomycin when VAN = 0 mg/mL. For experimental conditions 2 and 3, the cell culture medium was removed at the times specified in the table.

Time	Condition 1: Fixed Concentration of Vancomycin without Medium Removal	Condition 2: Fixed Concentration of Vancomycin with Medium Removal	Condition 3: Variable Concentration of Vancomycin
H0 to H1	VAN: 1X	VAN: 1X	VAN: 1X
H1 to H2		VAN: 1X	VAN = 0 mg/mL
H2 to H2.5		VAN: 1X	VAN: 1X
H2.5 to H3		VAN: 1X	VAN = 0 mg/mL
H3 to H4		VAN: 1X	VAN: 1X
H4 to H5		VAN: 1X	VAN = 0 mg/mL
H5 to H5.5		VAN: 1X	VAN: 1X
H5.5 to H6		VAN: 1X	VAN = 0 mg/mL
H6 to H6.5		VAN: 1X	VAN: 1X
H6.5 to H7		VAN: 1X	VAN: 3X
H7 to H7.5		VAN: 1X	VAN: 1X
H7.5 to H8.5		VAN: 1X	VAN = 0 mg/mL
H8.5 to H9		VAN: 1X	VAN: 1X
H9 to H9.5		VAN: 1X	VAN = 0 mg/mL
H9.5 to H10		VAN: 1X	VAN: 1X

## Data Availability

Not applicable.

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
