# Peer review of "Disturbance of Vancomycin Infusion Flow during Multidrug Infusion: Influence on Endothelial Cell Toxicity"

_antibiotics, 2021, doi:10.3390/antibiotics11010016_

Round 1
Reviewer 1 Report
Dear Authors
Thank you for providing us the opportunity to review your work.
Specific comments: Please consider rephrasing the first line of conclusion, one has to take two /three reads to understand what author is trying to convey.
Thank you
Author Response
Dear Doctor,
We thank you for this evaluation and your wise comment in order to improve our manuscript. We agree that the beginning of the conclusion is not clear enough.
We have now modified the first line of conclusion section:
“Although the endothelial toxicity of vancomycin is known to be concentration dependent, vancomycin did not induce excessive cytotoxicity beyond the cellular stress induced by the simulated multi-infusion conditions. These results can be explained by the cell stress-induced excess mortality associated with the frequent change of culture medium”.
(Initial sentence: “This study did not show endothelial cytotoxicity induced of variations in vancomycin concentration simulating multi-infusion in the same line, because variations in vancomycin concentration do not cause any more effect than that of a simple change in medium.”)
Reviewer 2 Report
This is a continuation of the authors’ research agenda into an important concern related to intravenous vancomycin administration. The manuscript is well written and the approach using human umbilical vein epithelial cells to simulate the toxicity of a vancomycin infusion in a clinical environment is novel and useful. I have a couple of suggestions that would improve an already strong submission.
Title: descriptive of the study
Abstract: well written summary
Key words: consider changing 'Endothelial Cells/drug effects' to 'human umbilical vein endothelial cells' and ' Intravenous/adverse effects' to 'infusions, intravenous.' Add 'simulation' and 'in vitro techniques.' Since you only studied only one adverse effect (an in vitro simulated phlebitis), I would not use a general term about adverse effects. Using MeSH terms will increase the manuscript's searchability.
Discussion: The authors discuss the potential for vancomycin leaching out of solution as a cause of phlebitis due to microparticles (Drouet M, Chai F, Barthélémy C, et al. Influence of vancomycin infusion methods on endothelial cell toxicity. Antimicrob Agents Chemother. 2015;59(2):930-934. doi:10.1128/AAC.03694-14), however, I think this has been covered well in other manuscripts, and doesn't need repeating here.
References are not in mdpi style. Please correct.
Well done. Thank you for the opportunity to review this interesting manuscript.
Author Response
Dear Doctor,
We thank you for this evaluation, for your helpful and wise comments in order to improve our manuscript.
We thank you for your comments on the chosen keywords. Indeed, our working question targeted the cutaneous toxicity of vancomycin and we modified the keywords according to your advice: Vancomycin; Phlebitis; Human Umbilical Vein Endothelial Cells; Infusions; Intravenous; Simulation; In Vitro Techniques; Toxicity Tests
(initial keywords: Vancomycin; Phlebitis; Endothelial Cells/drug effects; Intravenous/adverse effects).
Regarding the comments about the discussion section, we agree with you that the concentration-dependent toxicity of vancomycin is well known, and is described in the introduction section of this work and in the literature. A part of the sentence has been deleted as described below:
“As our previous study demonstrated that vancomycin endothelial toxicity was concentration-dependent, We hypothesized that vancomycin bolus induced by drug mass flow variations simulating clinical use could induce an excess death rate on endothelial cells surrounding the catheter outlet, where the endothelial cells are subjected to a continuous flow of vancomycin with poor dilution, due to the weak blood flow in a peripheral vein of patient in bed.”
We thank the reviewer tor his/her vigilance on the style of the references.
All references were checked and were in mdpi style.
Reviewer 3 Report
The authors investigated an influence of vancomycin flow to endothelial cell toxicity, and reported that vancomycin-induced phlebitis could have other causes than variation in drug mass flow. It is interesting and important research in clinical settings. However, themethods do not seem to simulate the clinical conditions, because the authors just removed and added the medium with vancomycin. Moreover, vancomycin continues to flow for 1 or 1.5 hours in clinical settings. The methods are that vancomycin remain in wells for several hours. The authors should perform suitable model on the clinical conditions, such as the chemostat.
- Introduction: The authors need to describe a frequency of vancomycin-induced phlebitis and an association between vancomycin concentration and the phlebitis in clinical settings.
- Method: Please categorize the methods using sub-title.
- Figure 1: The authors should add the control groups against each group; VAN at fixed concentration without and with medium removal, VAN with variable concentration.
Author Response
Dear Doctor,
We thank you for this evaluation and your wise comments in order to improve our manuscript.
To simulate the flow variations during multi-infusion, HUVECS cells were incubated with a variable concentration of vancomycin at different times. In order to analyze the effect of this frequent medium removal on monolayer cells, a control was performed where a fixed concentration of vancomycin solution was applied and removed at the same frequency. Although vancomycin infusion time is often 2 hours in North America, continuous infusion is preferred in Europe, hence our choice of 10-hour consecutive experiment. The analysis of cytoxicity after several hours of incubation allows us to quantify the toxicity of variations (high and low) in vancomycin levels induced by multi-infusion on a peripheral catheter. The results of this study allow us to consider other experiments more suitable for simulating a multi-infusion. This point is highlighted in the discussion section.
The sentence has been modified as follows:
1) "In this study, we wanted to mimic a 10-hour continuous infusion of vancomycin because continuous infusion of vancomycin is preferred in Europe. The cytoxicity analysis after several hours of incubation allows us to quantify the toxicity of variations (high and low) in vancomycin levels induced by multi-infusion on a peripheral catheter."
(initial sentence: "In this study, we wanted to mimic a 24h continuous infusion of vancomycin with multi-infusion on a peripheral catheter, commonly used in Europe in order to observe the impact of high and low vancomycin levels on endothelial toxicity.")
2) "However, our results showed that a variable concentration of vancomycin had no significant influence on toxicity compared to a fixed concentration of vancomycin with medium removal. This experiment did not induce excessive cytotoxicity beyond the cellular stress associated with frequent change of culture medium. The limitations of this study suggest that other experiments more suitable for simulating multi-infusion need to be performed to answer this scientific question.")
(initial sentence: "However, our results showed that a variable concentration of vancomycin had no significant influence on local toxicity compared to vancomycin at a fixed concentration with medium removal. We can suppose that the observed excess mortality was induced by cellular stress due to frequent change of culture medium and that the duration of the vancomycin bolus in our model was not significant enough to induce local toxicity.")
The answers are given below point by point:
1. We thank the reviewer for the comment about the introduction section and we agree that we didn’t emphasize enough the information about vancomycin-induced phlebitis.
Phlebitis is an adverse effect of vancomycin described in the literature with few specific details [1-2]. According to the summary of product characteristics, this is a commonly adverse effect occurring at a frequency of 1-10 % [3]. Children treated with vancomycin in Iran developed phlebitis at a frequency of 45.90 and 89.10%, depending on the infusion modality [4]. This treatment causes concentration-dependent phlebitis: it appears to be less frequent at a concentration below 5mg/mL and the administration is better tolerated at a final concentration of 2 mg/mL [4-5].
This notion has been added in the second sentence of the introduction section, with new references :
“Drug-associated phlebitis may be influenced by chemical and/or physical mechanisms. In particular, phlebitis is an adverse effect of vancomycin described in the literature [1-2], occurring at a frequency of 1-10 % according to the summary of product characteristics [3]. An Iranian study showed that the frequency of occurrence of this phlebitis in children treated with vancomycin was dependent on the infusion modality (45.90 and 89.10%) [4]. This treatment causes concentration-dependent phlebitis: it appears to be less frequent when administered at a concentration below 5mg/mL, and better tolerated at a final concentration of 2 mg/mL [4-5]. In previous studies, we analyzed factors influencing vancomycin (VAN)-associated phlebitis in peripheral intra-venous (PIV) infusion.”
[1] Peng, Y.; Li, C.Y.; Yang, Z.L.; Shi, W. Adverse reactions of vancomycin in humans: A protocol for meta-analysis. Medicine (Baltimore). 2020;99(38):e22376. doi:10.1097/MD.0000000000022376
[2] Bruniera, F.R.; Ferreira, F.M.; Saviolli, L.R.; Bacci, M.R.; Feder, D. ; da Luz Gonçalves Pedreira, M.; Sorgini Peterlini, M. A.; Azzalis, L. A.; Campos Junqueira, V. B.; Fonseca, F. L. The use of vancomycin with its therapeutic and adverse effects: a review. Eur Rev Med Pharmacol Sci. 2015;19(4):694-700.
[3] Agence nationale de sécurité du médicament et des produits de santé. Résumé des caractéristiques du produit vancomycine 500 mg poudre pour solution pour perfusion. Available online: http://agence-prd.ansm.sante.fr/php/ecodex/rcp/R0187161.htm (accessed on 17 December 2021).
[4] Tork-Torabi, M.; Namnabati, M.; Allameh, Z.; Talakoub, S. Vancomycin Infusion Methods on Phlebitis Prevention in Children. Iran J Nurs Midwifery Res. 2019;24(6):432-436. Published 2019 Nov 7. doi:10.4103/ijnmr.IJNMR_149_18
[5] Robibaro, B.; Vorbach, H.; Weigel, G.; Weihs, A.; Hlousek, M.; Presterl, E.; Georgopoulos, A.; Griesmacher, A.; Graninger, W. Endothelial cell compatibility of glycopeptide antibiotics for intravenous use. J Antimicrob Chemother. 1998;41(2):297-300. doi:10.1093/jac/41.2.297
(Initial sentence: “Drug-associated phlebitis may be influenced by chemical and/or physical mechanisms. In previous studies, we analyzed factors influencing vancomycin (VAN)-associated phlebitis in peripheral intra-venous (PIV) infusion.”)
2. We agree with the reviewer that the addition of sub-title in the method section will make this part better organized and understood. The added sub-titles are noted in red in the methods section copied below:
4.1. Cell culture
Used drugs, cell culture method, and the cell vitality assessment were described in our previous study Drouet et al.. The HUVEC cell line was provided by Promocell GmbH, Heidelberg, Germany. Briefly, these cells were cultured with cell culture medium (Endothelial cell growth medium enriched with endothelial cell growth SupplementMix, Promocell GmbH, Heidelberg, Germany) for 24h to obtain an 80% confluent monolayer in 96-well plate, as in vitro model of vascular endothelium.
4.3. Cytotoxicity testing
After a 24-hour culture with exposure to antibiotic solutions, cell reaction was evaluated by fluorometric assay with non-toxic AlamarBlue® dye (Interchim Montluçon, France), which is shown equivalent to the MTT (3-(4,5-dimethylthiazol-2-yl)-2,5-diphenyltetrazolium bromide) test to determine mammalian cell toxicity Hamid et al., « Comparison of Alamar Blue and MTT Assays for High Through-Put Screening »; Rampersad, « Multiple Applications of Alamar Blue as an Indicator of Metabolic Function and Cellular Health in Cell Viability Bioassays »..
After 24h of exposure, results were expressed as the percentage of cell viability compared to a control established with NaCl (0.9%) mixed with cell culture medium at 50/50 (v/v). Each test group was performed in triplicate.
4.2. Incubation with vancomycin
2.a. Methodology to study concentration-dependent cytotoxicity
As shown in previous study Drouet et al., « Influence of Vancomycin Infusion Methods on Endothelial Cell Toxicity »., vancomycin infusion at concentrations over 5 mg/mL induced 50% endothelial cell death over 24h, we, thus, only applied concentrations between 1 and 4 mg/mL.
The culture medium was removed from each well of monolayer cells, the cells were rinsed with Phosphate-Buffered Saline 1X then a vancomycin solution was added to the monolayer cells.
Firstly, to study The dose-dependent cytotoxicity of vancomycin, a range of vancomycin concentrations (1 – 4 mg/mL) was prepared with the vancomycin powder (Mylan, France) reconstituted in NaC1 (0.9%) to obtain a concentration of 500 mg/10mL, and diluted with cell culture medium at 50/50 (v/v).
2.b. Methodology to study multi-infusion-dependent cytotoxicity
Subsequently, to study the cytotoxicity of variations in vancomycin concentration by multi-infusion, Flow variations during multi-infusion therapy were simulated as follow : we set the sudden decrease in drug delivery by replacing vancomycin (VAN) with NaCl (0.9%) for 30 minutes or 1 hour every 1h30, and a sudden increase by trebling vancomycin concentration on one occasion for 30 min (Table 1, VAN with concentration variations). Schematic representation of variations in vancomycin concentration over 10h for the variable concentration group is shown in Figure 2.
To assess cell stress due to frequent medium removal from monolayer cells, we also set a control group, in which addition of vancomycin was replaced by new vancomycin solution at the same concentration at each hourly change (Table 1, VAN at fixed concentration with medium removal). After a 24-hour culture, the percentage of cell viability was compared to controls established with a fixed concentration solution of vancomycin with and without medium removal.
The excess cell death was calculated by comparing the cell death rate obtained with 1) vancomycin at fixed concentration without medium removal and 2) vancomycin at fixed concentration with medium removal.
4.4. Statistical analysis
Non-parametric tests were used to compare percentages of HUVEC surviving with the null hypothesis that there was no difference between the experimental conditions assessed. The Kruskal-Wallis test was used to assess the toxicity of vancomycin in the three conditions. In the presence of a significant p value (< 0.05), an analysis using the Conover and Iman method was performed to detect significant differences between couples of contact time. Each of these tests was performed with XLSTAT software version 2012.2.01 (Addinsoft, Paris, France).
3. We thank the reviewer for this important comment and we agree that the legend to Figure 1 did not identify the comparisons made.
The new legend is shown below:
Figure 1. Cell viability of HUVEC after 24h-contact with vancomycin. Vancomycin (VAN) concentration ranged from 1 mg/mL to 4 mg/mL. ** p<0.01, *** p<0.001, for the comparison between VAN at a fixed concentration without medium removal and VAN with variable concentration. n=3, error bars represent the Standard Error of the Mean.
Round 2
Reviewer 3 Report
The authors revised appropriately. No further correction is necessary.